# Electron Microscopy (EM) Analysis of Collagen Fibers in the Peri-Implant Soft Tissues around Two Different Abutments

**DOI:** 10.3390/jfb14090445

**Published:** 2023-08-29

**Authors:** Ugo Covani, Enrica Giammarinaro, Natalia Di Pietro, Simona Boncompagni, Giorgia Rastelli, Tea Romasco, Eugenio Velasco-Ortega, Alvaro Jimenez-Guerra, Giovanna Iezzi, Adriano Piattelli, Simone Marconcini

**Affiliations:** 1Department of Stomatology, Tuscan Dental Institute, 55041 Lido di Camaiore, Italy; covani@covani.it (U.C.); e.giammarinaro@gmail.com (E.G.); simosurg@gmail.com (S.M.); 2Department of Medical, Oral and Biotechnological Sciences, “G. d’Annunzio” University of Chieti-Pescara, 66100 Chieti, Italy; tea.romasco@unich.it (T.R.); gio.iezzi@unich.it (G.I.); 3Center for Advanced Studies and Technology—CAST, “G. d’Annunzio” University of Chieti-Pescara, 66100 Chieti, Italy; simona.boncompagni@unich.it (S.B.); giorgia.rastelli@unich.it (G.R.); 4Department of Neuroscience, Imaging and Clinical Sciences, “G. d’Annunzio” University of Chieti-Pescara, 66100 Chieti, Italy; 5Department of Stomatology, Faculty of Dentistry, University of Seville, 41013 Seville, Spain; evelasco@us.es (E.V.-O.); alopajanosas@hotmail.com (A.J.-G.); 6School of Dentistry, Saint Camillus International University of Health and Medical Sciences, 00131 Rome, Italy; apiattelli51@gmail.com; 7Facultad de Medicina, UCAM Universidad Católica San Antonio de Murcia, 30107 Murcia, Spain

**Keywords:** collagen fibers, concave abutment, healing abutment, peri-implant soft tissues, swine, ultrastructural analysis

## Abstract

The design of the implant prosthesis–abutment complex appears crucial for shaping healthy and stable peri-implant soft tissues. The aim of the present animal study was to compare two implants with different healing abutment geometries: a concave design (TEST) and a straight one (CTRL). Transmission electron microscopy (TEM) was used to quantify the three-dimensional topography and morphological properties of collagen at nanoscale resolution. 2 swine were included in the experiment and 6 implants per animal were randomly placed in the left or right hemimandible in either the physiologically mature bone present between the lower canine and first premolar or in the mandibular premolar area, within tooth extraction sites. Each CTRL implant was positioned across from its respective TEST implant on the other side of the jaw. After 12 weeks of healing, 8 specimens (4 CTRL and 4 TEST) were retrieved and prepared for histological and TEM analysis. The results showed a significantly higher percentage of area covered by collagen bundles and average bundle size in TEST implants, as well as a significant decrease in the number of longitudinally oriented bundles with respect to CTRL implants, which is potentially due to the larger size of TEST bundles. These data suggest that a concave transmucosal abutment design serves as a scaffold, favoring the deposition and growth of a well-organized peri-implant collagen structure over the implant platform in the early healing phase, also promoting the convergence of collagen fibers toward the abutment collar.

## 1. Introduction

The long-term success rate of implants depends on many factors, such as the accurate ex ante assessment of the patient’s local and systemic risk factors, the ideal implant positioning, the implant macro-geometry, the prosthetic rehabilitation, and the implant maintenance [1,2]. In addition to that, the peri-implant mucosal attachment, as well as acting as a physical barrier between the oral cavity and the osseous support of the implant, plays a key role in the prevention of microbiological infiltrations and inflammatory peri-implant diseases and contributes to the implant long-term success and survival [3]. Indeed, well-organized connective tissues around the implant were hypothesized to decrease early bone resorption by reducing inflammatory cell infiltration [4]. Accordingly, poor quality and quantity of peri-implant soft tissue could be associated with increased prosthetic failure over a long-term period [5]. Therefore, both the integrity of the epithelial lining and the health of the supra-crestal connective tissue are required to maintain implant health for a long time [6,7]. In 1996, it was established that mucosal thickness plays a crucial role in maintaining marginal bone stability, demonstrating that if the minimal requirement for the supracrestal tissue attachment (previously defined as biological width) [8], which includes a sufficient surface for both junctional epithelium and connective tissue attachments, is not met, bone resorption will take place [9]. After implant insertion, the healing period required for the formation and maturation of the supracrestal tissue attachment may last 6 to 12 weeks [10]. Even though the peri-implant soft tissue is created in response to surgical trauma or the implantation of a medical device, its dimension and composition have been constantly reported in different human histological studies [11]. On average, the supracrestal tissue attachment around implants including both the epithelial and the connective tissues measured as 3 to 4.5 mm [12].

Moreover, documenting the topography and the morphological properties of collagen fibers present around the implant neck could be essential to understanding how alterations in direction, periodicity, and diameter of collagen fibers could affect the biomechanical behavior of the peri-implant mucosa. Classical histological studies have described the arrangement of connective tissue fibers around implants in dogs and humans, attesting to the presence of parallel to long-axis, circular or ring-shaped, or inserted fibers [9,13]. Otherwise, other animal studies have described the presence of radial fibers, resembling dentogingival ones, especially around porous abutment surfaces [14].

Certain prosthetic abutments that underwent surface modifications have been able to generate a more robust and perpendicular connection between collagen fibers and the abutment. Notably, the presence of micro-grooves on implant collars produced using lasers proved high efficacy in promoting a seamless bond with the surrounding connective tissue on these surfaces [15]. The connection between the soft tissue and abutment surface provides a marked contrast to the migration of junctional epithelium toward the implant apex. This contrast contributes to the reduction of marginal bone loss (MBL) and leads to a substantial enhancement in the healing of both hard and soft tissues in the peri-implant area, as compared to using a machined surface. The same group [16] also reported that a laser-assisted new attachment procedure (LANAP) could induce regeneration of the periodontal tissues with the formation of cementum, periodontal ligament, and alveolar bone. These findings were supported also by Shapoff et al. [17] in a human study with the same laser microtextured abutments, in which it was reported optimal crestal bone levels, improved healing of the peri-implant soft tissues, and high tissue stability with a low depth of the sulcus.

In the literature, various factors have been reported to influence the quality and quantity of connective tissue attachment and healing around dental implants. For instance, different surface treatments, such as plasma or argon activation, air abrasion, acid etching, laser treatment, micro-grooving, and electrochemical oxidation, have been applied to achieve abutment micro-geometry and surface bio-activation [18,19], therefore influencing soft tissue morphogenesis. Additionally, a range of materials, including titanium, zirconium oxide, gold alloy, aluminum oxide, ceramics, titanium nitride, and hydroxyapatite, have been utilized for the same purpose. Notably, titanium and zirconia have demonstrated favorable soft tissue responses, while the use of gold alloy failed to establish an appropriate peri-implant soft tissue response [14,20,21]. Overall, rougher surfaces have exhibited improved peri-implant soft tissue characteristics, and it has been observed that epithelial cells adhere more effectively to metallic surfaces compared to ceramic surfaces [21].

Moreover, the abutment design was also demonstrated to affect the peri-implant soft tissue biological response. Collagen fibers, indeed, are not predetermined, yet they depend on the local environment. Rodriguez and co-workers [22] reported that around implants with a platform-switching design, the circular orientation of collagen fibers was observed as the main arrangement in a cross-sectional view. They argued that by increasing the room for soft tissues by changing the abutment design or its transversal discrepancy with respect to the implant platform, the supracrestal connective tissue fibers would be retained in a stable coronal position.

The geometry and behavior of the pre-existing extracellular matrix (ECM) might be modulated using mechanical and geometrical cues [23]. The design features of the implant prosthesis–abutment complex have been proven to be crucial for shaping healthy and stable peri-implant soft tissues [24]. Over the years, different abutment shapes have been proposed, from scalloped, parallel-walled, and platform-switching designs to concave ones [25,26,27]. These latter present an inward narrowed profile that creates a macroscopic concave profile just above the implant platform [28]. Lately, several authors have suggested that the concave design provides more space for the formation of a stable blood clot, further promoting fibroblast proliferation and migration, ECM deposition and protein adsorption, granulation tissue formation, ECM remodeling, and an increased contact area between soft tissues and the abutment, all leading to greater connective tissue stability and mechanical properties [3,14,29,30]. In this regard, Rompen et al. [31] demonstrated that a concave transmucosal design determined improved soft tissue stability with respect to divergent transmucosal abutments. In animal experimental studies, instead, other authors found denser and better-organized collagen fibers with higher connective tissue attachment, as well as significantly less peri-implant bone resorption around concave abutments, also when compared with straight designs [32,33]. Nonetheless, in their animal study, Delgado-Ruiz et al. [26] reported a lower thickness of the peri-implant soft tissues around a concave geometry of the abutment.

Considering all the above, here it is hypothesized that a concave implant neck might trigger spontaneous alignment of the collagenous network, therefore affecting fibroblast polarization, migration, and fiber growth direction and arrangement. To confirm this hypothesis, it was decided to perform a proof-of-principle animal study to study the structure and distribution of collagen fibers and bundles in the peri-implant soft tissues by comparing two implants with identical bodies but different healing abutment geometries: the Test one presented a 2 mm concave area above the implant platform, chosen according to the positive results reported in a histological animal study and a clinical study with a similar concave profile [3,31], whereas the Control abutment had a parallel-walled healing screw.

## 2. Materials and Methods

### 2.1. Implant Characteristics

All implants were tapered shaped (IK Internal Hexagon, RESISTA^®^ Company, Ing. Carlo Alberto Issoglio & C. S.r.l., Omegna, Italy). Test abutments (TEST) presented a 2 mm height concave portion with a double acid-etched (DAE) surface, whereas Control ones (CTRL) were parallel-walled shaped with a DAE surface. Both presented a 4 mm diameter switching platform and a length of 10 mm (Figure 1).

### 2.2. Ethical Statement

This animal study was approved by the Animal Ethical Committee of the Junta de Andalucia, Consejeria de Agricultura, Ganaderia, Pesca y Desarollo Sostenible on 14 December 2021 (n° 29/11/2021/184). The animal study and procedures were performed in accordance with Spain’s animal protection laws and according to the Animal Research: Reporting of in Vivo Experiments (ARRIVE) guidelines [34] in a randomized prospective design.

### 2.3. Experimental Animals and Housing

2 swine (sus scrofa), aged on average 3 years old, were acclimated for three weeks before the initiation of the study. The two animals were identified using an ear tag. An antibiotic-free diet, softened by soaking in water, was provided. Water was available ad libitum. The person in charge of animal welfare took care of aeration and food and water administration, as well as animal behavioral and health conditions throughout the study period. The whole study was accompanied and monitored by a veterinarian, and surgeons with extensive experience performed all surgical procedures.

### 2.4. Experimental Design

Animals had implants placed in the left or right mandibular alveolar ridges. Implants were either placed in the physiologically mature bone present between the lower canine and first premolar or at the mandibular premolar area, within tooth extraction sites. All implants received a healing abutment at the time of placement and 12 weeks of healing were allowed. Each animal received 6 implants, 3 per hemimandible. CTRL and TEST implants were positioned across the jaw in a symmetrical and well-controlled manner. A total of 12 weeks after the implant placement, all animals were euthanized. Therefore, a total of 12 implants were placed. 2 CTRL implants and 2 TEST implants were excluded from further analysis because of early implant failure. In the end, a total of 8 implants were analyzed (CTRL, n = 4 and TEST, n = 4).

### 2.5. Surgical and Terminal Procedures

Before surgical intervention, animals were fasted overnight and weighed. On the day of surgery, all animals were anesthetized with intramuscular (IM) medetomidina 0.05 mg/kg + Zoletil (zolacepam + tiletamina) 3 mg/kg.

After that, a mask inhalation of 2–5% of Isoflurane mixed with oxygen was administered. Animals were transferred to the surgical area and intubated with an endotracheal tube, after which general anesthesia continued with 2–5% of Isoflurane. Monitoring of heart rate, blood oxygen saturation, and blood pressure occurred during the entirety of the procedures, as well as the post-operative period. 

All surgical procedures were performed under aseptic conditions in an animal operating theater under general anesthesia. Tooth extraction was carefully completed: for all teeth, gentle pressure was applied to the gingival sulcus using a small periosteal elevator, after which mandibular premolars and molars were sectioned in a buccolingual direction at the furcation between the mesial and distal root. A rotary instrument was used for sectioning; then, a straight elevator was used to confirm sectioning. After that, the mesial and distal roots were elevated and removed using dental forceps.

In mature sites, a No. 15c blade was used to create a midcrestal incision in the area between the canine and the first premolar. A full-thickness mucoperiosteal flap was elevated and implants were placed at least 1.5 mm apart from the neighboring teeth and housed within the buccal and lingual plates using manufacturer guidelines for drilling protocol. Healing abutments were placed, and the site was closed with 4–0 silk sutures. All implants were placed equicrestally, and no bone graft was placed.

Within the first days after surgery, all animals were monitored routinely, and further analgesia was given if necessary. Post-operative surgical pain was relieved using 0.12–0.24 mg/kg buprenorphine HCl, administered subcutaneously (SC). Animals were sacrificed 12 weeks after surgery. Following sedation using the aforementioned agents, cardiac arrest was induced by administration of 110 mg/kg Pentobarbital intravenously (IV) at each previously mentioned timepoint.

### 2.6. Samples Preparation

Block sections were retrieved using an oscillating autopsy saw to keep the soft tissue intact. Samples of peri-implant soft tissues for histological analysis were fixed by immersion in 10% buffered formalin, dehydrated in increasing series of alcoholic rinses, and finally embedded in glycol-methacrylate resin (Technovit 7200 VLC, Wehrheim, Germany). The specimens were processed according to the protocol described in a previous study by Iezzi and collaborators [35]. Briefly, they were sectioned along its longitudinal axis to obtain histological longitudinal sections of the peri-implant tissues. Histological analysis was carried out under a light microscope (Laborlux S, Wetzlar, Germany) connected to a high-resolution video camera (3CCD, JVCKY-F55B, JVC, Yokohama, Japan) and interfaced with a PC. 

Samples of peri-implant soft tissues for transmission electron microscopy (TEM)
analysis, instead, were preserved in 3.5% glutaraldehyde in 0.1 M sodium cacodylate (NaCaCO) buffer.

### 2.7. Electron Microscopy (EM)

#### 2.7.1. Preparation and Analysis of Samples for EM

All implants and associated adjacent peri-implant soft tissues were removed from the mandible of each swine. 4 specimens were retrieved around parallel-walled abutments (CTRL) and 4 specimens around concave abutments (TEST) and fixed at room temperature (RT) with 3.5% glutaraldehyde in 0.1 M NaCaCO buffer (pH 7.2) and stored at 4 °C in the fixative until shipment. Small portions of fixed soft tissues carefully dissected from the area around the implant were rinsed in 0.1 M NaCaCO buffer and then, post-fixed 2% osmium tetroxide (OsO_4_) in the same buffer for 1 h, block-stained with saturated uranyl acetate, rapidly dehydrated in graded ethanol and acetone, and embedded in epoxy resin (Epon 812) [36]. For electron microscopy (EM), ultrathin sections (~40 nm) were cut in a Leica Ultracut R microtome (Leica Microsystem, Wetzlar, Germany), using a Diatome diamond knife (Diatome Ltd., Biel, Switzerland), and after double staining with uranyl acetate and lead citrate, they were examined at 60 kV with an FP 505 Morgagni Series 268D electron microscope (FEI Company, Brno, Czech Republic), equipped with a Megaview III digital camera (Olympus, Tokyo, Japan) and Soft Imaging System (GmbH, Munster, Germany).

#### 2.7.2. EM Ultrastructural Analysis of Collagen Bundles

For EM qualitative and quantitative analysis small samples were taken from a ring of tissue dissected all around the area of the abutment. The ultrastructural analysis of collagen from peri-implant soft tissues was mostly performed in images showing the cross-sectional appearance of the collagen fibers, taken from longitudinal sections of the tissues. Only sample regions near the abutment surface (CTRL, n = 4 and TEST, n = 4) were observed.

#### 2.7.3. EM Quantitative Analysis of Collagen Bundles

For quantitative analysis, 15 micrographs/group were randomly collected from non-overlapping regions at 7100× of magnification and used for the following quantitative analysis:(i)In each micrograph, the total area covered by collagen bundles was evaluated by drawing the outline of each bundle using the Soft Imaging System (GmbH, Muenster, Germany). All measured bundle values were then mathematically summarized. Only cross-sectioned bundles with a minimum size of 0.5 µm^2^, where collagen fibers were distinguishable and not longitudinally oriented, were considered for the analysis. Considering that each micrograph at 7100× of magnification covers 142.6 µm^2^ of sample, the relative presence of collagen bundles (%) in each sample was obtained by dividing the number of total outlined collagen fibers (in µm^2^) by the total area of analyzed samples (i.e., 142.6 µm^2^ × 15 micrographs).(ii)In each micrograph, the number of longitudinally oriented bundles of collagen (of different sizes) was counted and reported as mean ± standard error of the mean (SEM) in 100 µm^2^.

### 2.8. Statistical Analysis

In all comparisons performed between CTRL and TEST conditions, not normally distributed data were found and analyzed using a non-parametric *t*-test (Mann–Whitney test). The experimental values were elaborated using the statistical software package GraphPad Prism Software Analysis version 9.0 (San Diego, CA, USA), and the statistical significance of the differences between the groups was determined for a *p* < 0.05. Data were expressed as the mean ± SEM or standard deviation (SD).

## 3. Results

### 3.1. Radiographic Analysis

After 12 weeks, proper histologic healing was observed around both CTRL and TEST implants (Figure 2). The radiographic exams also revealed a good osseointegration of implants.

### 3.2. Histological Analysis

For histological analysis, longitudinal undecalcified sections were obtained. Histological results (Figure 3) showed the presence of peri-implant soft tissues in close connection with both TEST and CTRL abutments. However, although this is only a qualitative finding, the soft tissue appears more adherent to the concave abutment than the straight one.

### 3.3. EM Ultrastructural Analysis of Collagen Fibers in the Peri-Implant Soft Tissue

Ultrastructural analysis of the peri-implant soft tissue taken from sites near the two different implants (CTRL and TEST) was initially performed blinded. At the EM analysis, peri-implant soft tissues were primarily constituted by collagen fiber bundles and cells, i.e., fibroblasts (Figure 4A, f). Collagen fibers appeared as several long, parallel, and straight tubules so that when cut transversally (i.e., in longitudinal images), they appear as “bunches of circular spots” (Figure 4, insets), indicating bundles of heterogeneous size. In longitudinal images, collagen fibers had a quite uniform diameter (0.125 nm) in both CTRL and TEST samples (Figure 4, insets).

Cell populations of the peri-implant soft tissues were mostly constituted by fibroblasts (Figure 4, f), which exhibited a stellate appearance (note how fibroblast processes segregated individual collagen bundles, Figure 4, black arrows). Interestingly, during the EM analysis of the cross-sectioned collagen bundles at low magnification images (7.1k), the presence of different structural arrangements of collagen fibers between samples was quite evident. Specifically, comparing the different appearance of collagen distribution and organization allowed us to divide samples into 2 groups: CTRL specimens, in which an extensive aggregation of thick collagen bundles was rare or absent (Figure 4A), and TEST samples containing a high-density large aggregation of tightly packed and sorted collagen fiber bundles (Figure 4B). With more careful analysis, we also observed that in CTRL samples, there were only a few assembled collagen fibers, forming small scattered bundles, while in TEST samples, the collagen bundles were notably thick and dense, typically covering the entire area of the analyzed section (Figure 4). Specifically, in CTRL samples (Figure 4A), the collagen matrix was composed of scattered collagen bundles randomly distributed in the extracellular space at variable distances from each other. On the contrary, large areas were observed in TEST samples, where very thick collagen bundles were densely packed with each other without leaving much space for the extracellular material (Figure 4B). These data, for the first time, suggest a different growth and assembly of the collagen matrix around TEST abutments when compared to CTRL abutments. The concave shape seemed to determine an increased bundle size of collagen fibers.

### 3.4. EM Quantitative Analysis of Collagen Fiber Bundles

To confirm the qualitative results, a quantitative EM analysis of images was performed (Figure 5). In detail, from longitudinal CTRL and TEST images of the area of interest, the following features were evaluated: (i) the percentage of the total analyzed surface area covered by cross-sectioned collagen fibers (Figure 5E); (ii) the average size of the same collagen bundles (Figure 5F).

To better allow the visualization of collagen bundles’ size, their surfaces have been highlighted with light green in the longitudinal images (Figure 5B,D). The mathematical sum of each value gave a representative percentage of the surface area covered by collagen and the results have been numerically reported in Figure 5B,D. Quantitative analysis of the total surface area covered by collagen fibers indicated that the use of TEST implants was quite effective in aiding the formation and aggregation of collagen bundles in larger areas than with CTRL ones. Notably, the percentage of total surface covered by collagen was significantly higher in TEST samples (approximately 47%) with respect to CTRL ones (about 18%) (Figure 5E and Table 1).

Furthermore, the use of TEST abutments was also effective in significantly increasing the average size of the collagen bundles, from 4 µm^2^ of the CTRL samples to 13 µm^2^ (Figure 5F, Table 2, and Figure 6). The number of longitudinally oriented collagen bundles per 100 µm^2^ was lower in TEST samples than in CTRL samples (Figure 5B,D).

### 3.5. Additional Ultrastructural Observations

In addition to the qualitative and quantitative differences described so far, other distinctions have been found between CTRL and TEST specimens. Indeed, it was possible to note the presence of areas characterized by abrupt changes in collagen bundle direction. In longitudinal images of peri-implant soft tissues, collagen fibers usually appeared as small circles closely assembled in bundles of heterogeneous sizes (Figure 4 and Figure 5). However, the presence of collagen bundles with longitudinally oriented fibers was occasionally observed (Figure 7).

After a careful examination of the specimens, longitudinally oriented fibers appeared different between CTRL and TEST samples (Figure 7). In particular, in CTRL specimens (Figure 7A,B), longitudinally oriented collagen fibers (L) are usually assembled in small-sized bundles and involve collagen fibers with a quite random orientation between each other. In TEST specimens, instead, longitudinally oriented collagen fibers (L) are assembled in larger bundles involving several, straight, and parallel-oriented fibers (Figure 7C). The number of longitudinally oriented collagen bundles per 100 µm^2^ was quantified, and indeed it was found that in CTRL samples their incidence was significantly higher than in TEST samples (Table 3).

## 4. Discussion

The present study aimed to compare the peri-implant soft tissue healing process around non-submerged implants with a parallel-walled abutment or with a concave abutment inserted in a swine model.

In a study conducted by Berglundh and Lindhe in 1996 on an animal model [9], they revealed that a specific level of mucosal thickness is essential for the formation of the supracrestal tissue attachment around dental implants. In the case of deficiency, crestal bone resorption will take place until enough space is created to accommodate both connective tissue and junctional epithelium. Despite their similarity in composition and structure, research has indicated that this attachment apparatus is longer around dental implants when compared to natural dentition, therefore necessitating a greater amount of soft tissue height around implant fixtures [13,37,38].

During recent years, several animal and human reports have described the characteristics, arrangement, and structure of peri-implant soft tissues using different techniques such as light microscopy, polarized light microscopy, scanning electron microscopy (SEM), TEM, and high-resolution X-ray phase-contrast micro-topography (XPCT) [14,26,35,39]. As an example, in 2 animal studies performed more than 30 years ago in monkeys [14,40], it was found that large collagen fiber bundles ran around the implant collar in a parallel way, according to a tangential circular arrangement and converging to form a “circular ring”. TEM findings further showed that these circular fibers appeared to be constituted by bundles of parallel collagen fibrils with a mean diameter of 90 nm, but the inner bundles running close to the metal surface presented a less regular arrangement; indeed, they had a random course, as well as thinner and different diameters with a mean of 45 nm. Contrarily, Iezzi et al. in 2021 [35] showed transverse and longitudinal intertwined collagen bundles in a high-resolution XPCT study of peri-implant tissues around human retrieved implants. When evaluating the longitudinal sections, it was found that the closer the fiber bundles were to the metal surface, the more symmetric and regular their direction was. On the other hand, when analyzing transverse bundles of collagen fibers, it was seen a semicircular direction of these bundles, so fibers ran around the abutment, following its circular profile. Similar results were also reported by other researchers. For instance, in an animal study conducted by Bolle et al. [41], it was found that collagen fibers ran medially toward the healing abutment in a perpendicular direction and the connective tissue was dense, rich in fibroblasts and collagen fibers, which were parallel to the implant surface. Other swine studies [42] reported that in some areas, the connective tissue was well organized, while in others, the fibers exhibited a lack of organization, displaying an ambiguous and indistinct orientation. Furthermore, in human studies [35], a three-dimensional (3D) network of collagen fibers was reported around Cone–Morse implant connections. Similar results were reported by Mangano et al. [43] using the polarized light and SEM. Collagen fibers were oriented perpendicularly up to a distance of 100 µm from the implant surface, where they became a dense and chaotic 3D network of parallel fibers running in different directions and an intimate contact of the fibrous matrix with the implant surface was found. After maturation, peri-implant connective tissue had scarce cellularity and blood vessels but became rich in collagen fibers with a few scattered fibroblasts [14].

This structure of the connective tissue has been reported to play a relevant role in the prevention of epithelium down growth and in offering mechanical protection to the osseointegrated part of the implant [35]. Also, the dense 3D framework of the connective tissue bundles determines the mechanical resistance of soft tissues to withstand forces produced during chewing [44]. There is a significant correlation between the degree of fiber orientation in the tissue and its mechanical parameters, such as the elastic modulus.

The present results demonstrated that the introduction of a concave profile in the abutment could lead to the organization of a strong wire-shaped connective tissue cuff (about 0.5 mm of thickness) over the implant platform, in which cells, fibrils, and left ECM presented a high degree of anisotropy. In this way, it has been shown how it is possible to modulate dimensions and the quality of fibers, as well as the morphogenesis of a highly aligned capillary-like network, by controlling the spatial organization of the neo-formed ECM. Taken together, these data suggested that during ECM maturation around the abutment interface, the local microenvironment could be influenced by the macroscale tissue geometry, which may trigger long-range signals by inducing internal gradients of mechanical cues, as already reported in the literature [23]. Therefore, tissue geometry acts as both a template and an instructive cue for further morphogenesis. In the present study, the CTRL group with a parallel-walled neck showed a significantly greater ratio of randomly distributed fibers. However, it is well known that moderate crosslinking is beneficial to the mechanical properties of collagen fibers, but excessive crosslinking leads collagen fibers to become more fragile [45]. In the TEST concave group, instead, collagen fibers appeared to be organized in abundant parallel bundles when seen in cross-section and so running circumferentially around the implant when seen from above/axial planes. This result is in line with previous literature describing collagen fiber orientation around implants with a switching platform interface, considered to be an additional “mechanical retention factor” for periodontal fiber orientation [22,25] Similarly, studies conducted on other animal models, including monkeys and dogs, have shown a supracrestal circular collagen fiber network that is even comparable to gingival ligaments [14,22].

Overall, it can be argued that the mechanical environment could play an extremely important role in collagen fiber orientation. It is believed that this phenomenon is caused by an uneven surface shear that gradually attenuated its effect with the distance [46]. The geometry of the artificial substrate might provide contact guidance for the formation of a highly polarized capillary-like network, suggesting clinical applications in triggering fast angiogenesis and perfusion in wounded tissues around the implant [47]. Specifically, collagen fibers can remodel into aligned, anisotropic ensembles under mechanical stimuli, orienting fibers into the direction of the highest applied strain. Specifically, collagen self-assembly is an entropy-driven process caused by the loss of water between monomers [48]. The goal of the mechanics-mediated fiber orientation experiments is not to recombine the collagen monomer by overcoming the interaction between monomers but to impose additional external forces on the interacting collagen monomer based on the intermolecular interaction, which can lead monomers to bond along the force direction. It can be speculated that, when the distal part of fibers meets the curve perimeter of the abutment concavity, the cellular contraction can generate sufficient force to trigger the aggregation of fibers into bundles.

Other authors believed that the organization of collagen fibers would be mainly dependent on function, namely implant loading [49]. This would lead to the interpretation of radial fibers as a circular ligament around implants. Also, the same authors have demonstrated that this collagen cuff appears to be linked to the periosteum by means of oblique bundles. However, there are no time-dependent studies demonstrating this assumption, nor studies assessing the arrangement of collagen in different rehabilitation designs. In addition, one hypothesis would not exclude the other and vice versa.

It must be remarked that collagen is a well-engineered molecule with native weak points that represent the binding sites for metalloproteinases (MMPs) and bacterial collagenases, a mechanism favoring the regulation of collagen reshaping upon precise stimuli. It has been demonstrated that strain and external loading on fibrils could reinforce collagen in the direction of loading and inhibit the spontaneous formation of entry points for MMPs, therefore limiting their accessibility and collagen degradation [50]. It follows that directionality and immediate tension on the early wound around implants might control collagen assembly and maturation. Many human tissues are featured by specific alignment patterns involving the ECM of the interstitial connective tissue, stromal cells, and vascular network [51]. Collagen arranged in bundles of aligned fibers controls not only the mechanical properties of tissues, but its density and alignment direction also triggers the polarization of several biological phenomena: cell migration, morphogenesis, vascularization, innervation, tissue regeneration, and wound healing [52]. In any case, having the possibility to control the alignment of a fibroblast-synthetized ECM network still represents a challenge in dentistry.

## 5. Conclusions

In conclusion, within the limitations of the present study due to the use of a small number of animals and implants that might bring uncertainty and risk to the research results, the present study on the peri-implant connective tissue structure evaluated by histological and TEM analysis showed that the concave transmucosal design could favor the deposition and growth of the connective tissue. This concavity generated a significant amount of connective tissue in the early healing phase, increased the thickness of this circular peri-implant network, and promoted the convergence of collagen fibers toward the abutment collar with the formation of a wide circular collagen structure over the implant platform.

Therefore, starting from our proof-of-principle animal study, future research involving a larger number of animals and implants, as well as using other mechanical detection methods together with histological and TEM analysis, will be necessary to confirm and strengthen the present results.

## Figures and Tables

**Figure 1 jfb-14-00445-f001:**
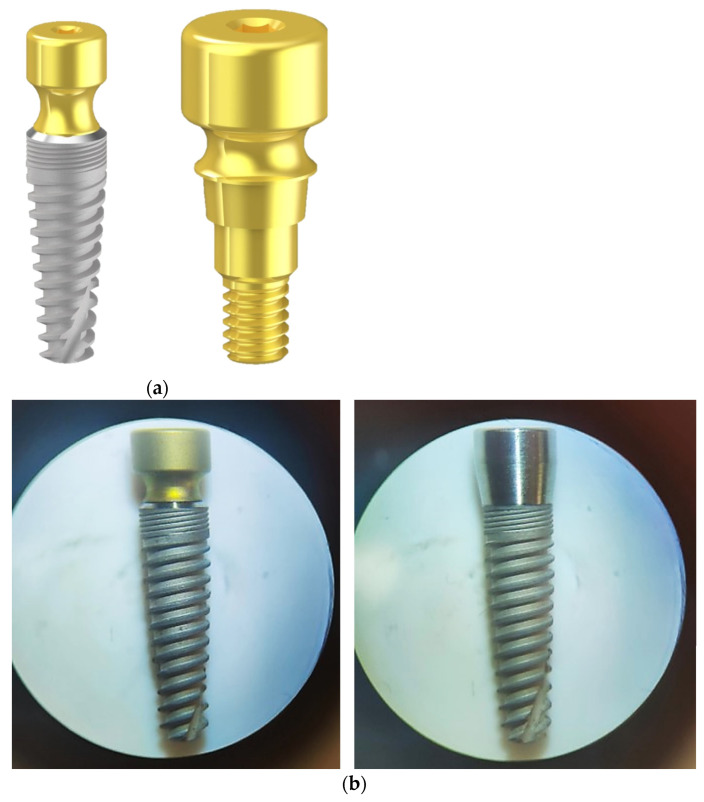
(**a**) Test implant and abutment (TEST); (**b**) Optical microscopy images of TEST implant (on the left) and Control implant (CTRL, on the right).

**Figure 2 jfb-14-00445-f002:**
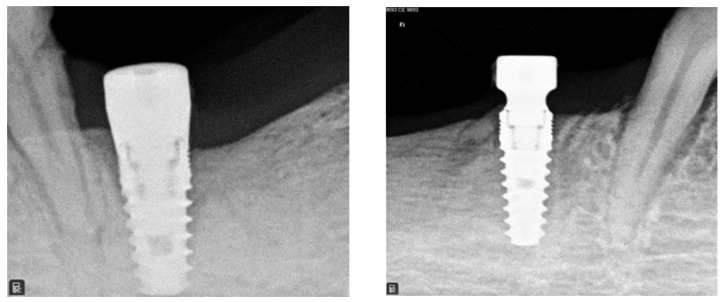
Representative intraoral radiographs presenting 2 implants placed in extractive sites of 2 premolars at the 3-month follow-up visit. A CTRL implant with a parallel-walled abutment on the left, and a TEST implant with a concave abutment on the right.

**Figure 3 jfb-14-00445-f003:**
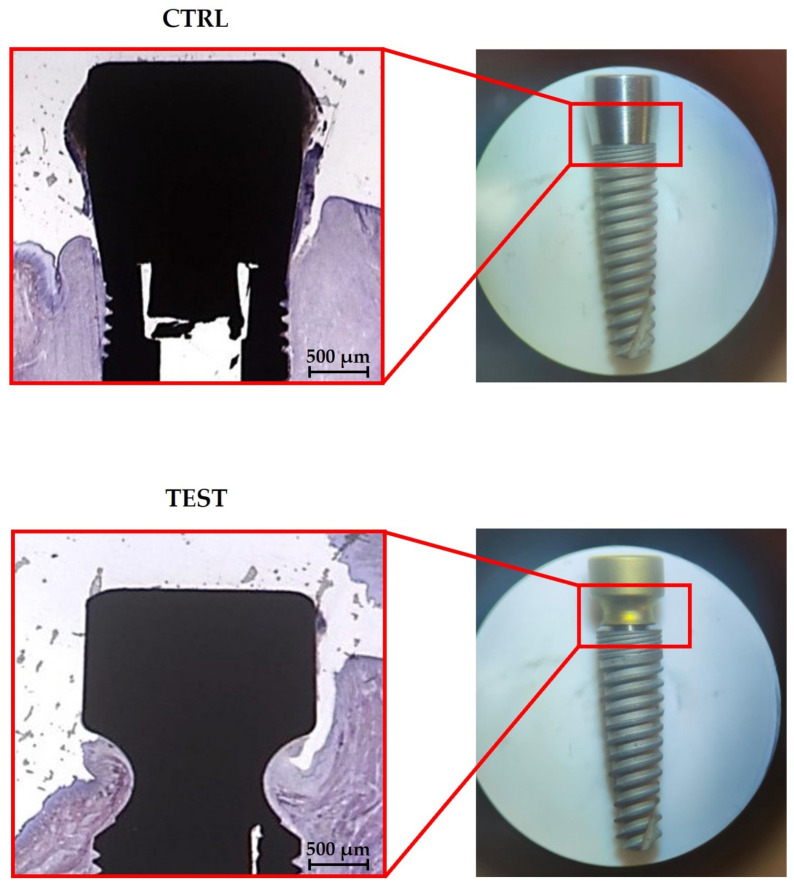
Histological longitudinal sections of the implant–abutment units. In the upper part, soft tissues surround the CTRL abutment. In the lower part, soft tissues surround the TEST abutment. (Acid fuchsin-Toluidine blue 20×).

**Figure 4 jfb-14-00445-f004:**
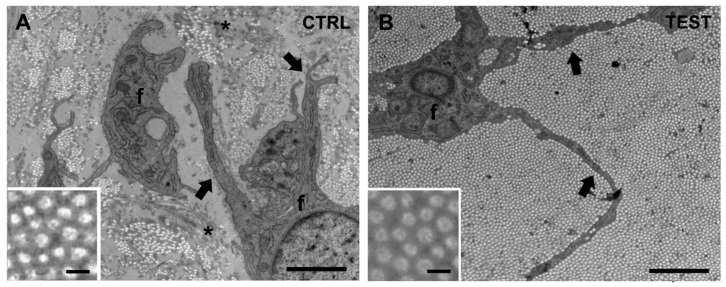
Representative electron microscopy (EM) images of peri-implant soft tissues around the 2 abutments. The peri-implant soft tissue is primarily constituted by collagen fiber bundles and fibroblasts (f); (**A**) in CTRL samples, only a few collagen fibers assembled, forming small, scattered bundles, compared to TEST samples; (**B**) in TEST samples, collagen bundles were thick and dense, often covering a large area of the analyzed section. NOTE: In the analyzed area, collagen fibers are mainly “cross-sectioned” and appear as small circles (insets), assembling in bundles of different sizes. Asterisks in panel (**A**) refer to longitudinally oriented collagen fiber bundles. Black arrows point to fibroblast processes (f). Scale bars: 2 µm; insets, 0.1 µm.

**Figure 5 jfb-14-00445-f005:**
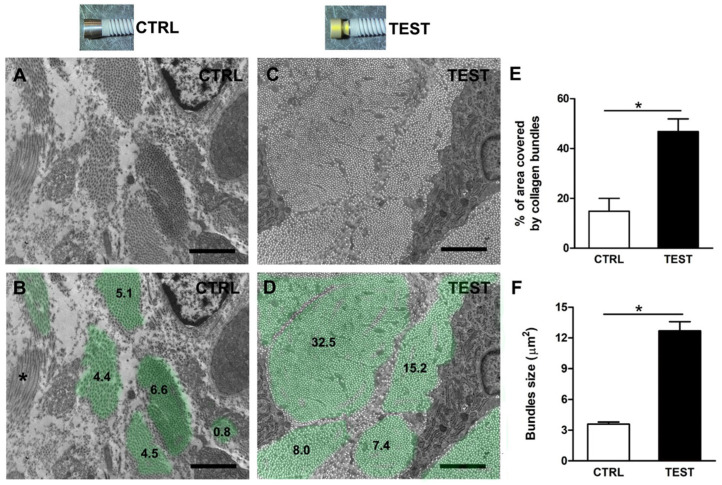
EM quantitative analysis of cross-sectioned collagen fiber bundles from CTRL and TEST samples. Representative EM longitudinal images of collagen fiber bundles around (**A**) CTRL and (**C**) TEST samples, (**B**,**D**) and corresponding collagen bundles’ surfaces are highlighted in light green. Numbers refer to bundles’ surface areas in µm^2^. Asterisk in panel (**B**) (*****) refers to a longitudinal collagen fiber bundle; (**E**) Bar plot showing the quantitative analysis of the percentage of the analyzed area covered by collagen fibers; (**F**) Average size of the collagen bundles. Scale bars: 2 µm. * *p* < 0.05.

**Figure 6 jfb-14-00445-f006:**
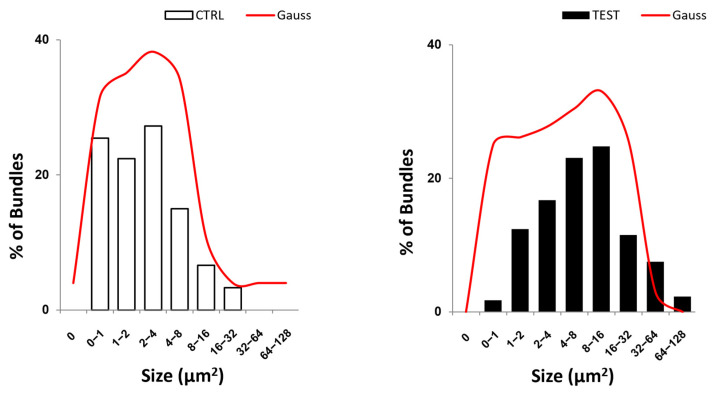
Representative bar and curve plots displaying the distribution frequency of bundle size for both CTRL and TEST groups. The analysis of the distribution frequency of cross-sectional area, i.e., size of bundles from CTRL and TEST groups, revealed that most bundles in the CTRL group have an average value of 3.4 ± 4.0, while in the TEST samples, the average is significantly increased to a value of 12.8 ± 16.0. This is also demonstrated by the leftward shift of the CTRL frequency distribution curve compared to the TEST curve.

**Figure 7 jfb-14-00445-f007:**
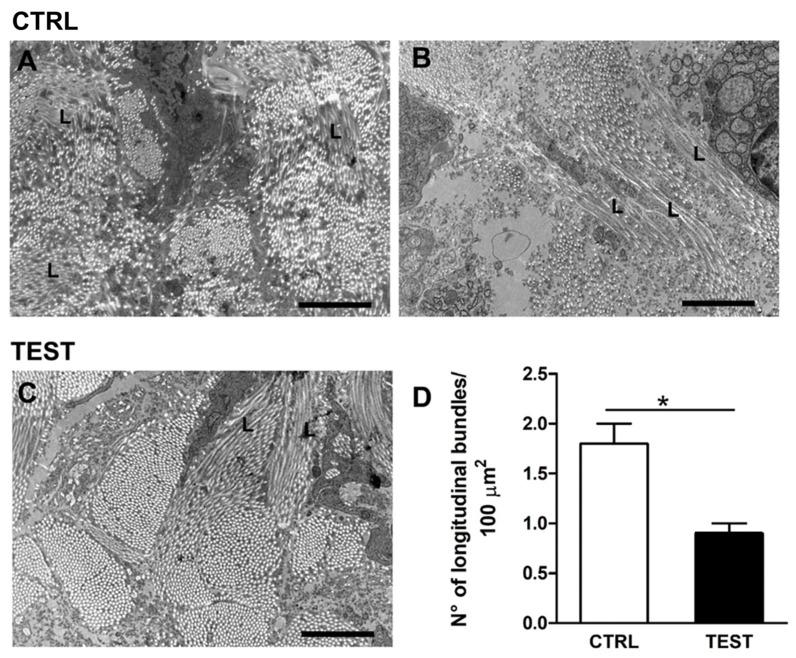
Representative EM images of different collagen fibers’ orientation and relative quantification in (**A**,**B**) CTRL and (**C**) TEST samples. In peri-implant soft tissue of the longitudinally analyzed areas, collagen fibers mostly appeared as described in Figure 4, i.e., as delimited circular spots of different sizes. Occasionally, but more frequently in CTRL than in TEST samples, a longitudinal orientation of collagen fibers (L) was present; (**D**) Bar plot showing the average number of longitudinal fiber bundles per 100 µm^2^ in CTRL and TEST samples. Scale bars: 2 µm. * *p* < 0.05.

**Table 1 jfb-14-00445-t001:** Percentages of the analyzed area covered by collagen bundles. Data are shown as mean ± standard deviation (SD) (* *p* < 0.05 vs CTRL).

	Sample 1	Sample 2	Sample 3	Sample 4
CTRL	18 ± 5	28 ± 3	8 ± 3	5 ± 1
TEST	47 * ± 11	46 * ± 6	63 * ± 10	43 * ± 19

**Table 2 jfb-14-00445-t002:** Bundles size (µm^2^). Data are shown as mean ± SD (* *p*< 0.05 vs CTRL).

	Sample 1	Sample 2	Sample 3	Sample 4
CTRL	4.2 ± 5.5	4.7 ± 4.4	2.6 ± 3.8	1.5 ± 1.3
TEST	20.7 ± 27.7 *	7.5 ± 9.1	23.7 ± 12.6 *	12.6 ± 16.6 *

**Table 3 jfb-14-00445-t003:** Number of longitudinal bundles/100 µm^2^. Data are shown as mean ± SEM (* *p* < 0.05 vs. CTRL).

	Sample 1	Sample 2	Sample 3	Sample 4
CTRL	1.2 ± 0.3	0.9 ± 0.2	3.4 ± 0.4	1.7 ± 0.2
TEST	1.1 ± 0.2	0.8 ± 0.2	1.1 ± 0.3 *	0.5 ± 0.2 *

## Data Availability

Data are contained within the article and available on request from the corresponding author. The data are not publicly available due to privacy.

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
