# Peer review of "Electron Microscopy (EM) Analysis of Collagen Fibers in the Peri-Implant Soft Tissues around Two Different Abutments"

_jfb, 2023, doi:10.3390/jfb14090445_

Round 1

Reviewer 1 Report

The study question and design are important in the context of the field. However, the study evaluating the design of the implant prosthesis-abutment complex reports insufficient information to the reader on factors that may influence the quality and quantity of connective tissue attachment around dental implants. in fact, not only the design of the abutment, but also types of abutments materials (titanium, ZrO2 and Au/Pt-alloy) have been shown to influence the peri-implant soft tissue healing. Moreover, several studies have shown that also the microgeometry of the abutment (regular smooth surface, textured surface, micro-grooved surface, etc) is able to influence the peri-implant connective tissue attachment quality. The reviewer suggests that the authors report, albeit summarily, the studies and conclusions present in the literature on all the factors that can influence the peri-implant connective tissue attachment. 

Abstract: 

37-40 “These data suggested that a grooved transmucosal abutment design serves as a scaffold, favoring the deposition and growth of a well-organized peri-implant collagen structure over the implant platform in the early healing phase, also promoting the convergence of collagen fibers toward the abutment collar. 

The word "groove" is not relevant to the material used as a test and is misleading to the reader. The reviewer suggests that the word "concave/ concave profile" be used throughout the manuscript.

Introduction 

56-57 “After implant insertion, the healing period required for the formation and maturation 56 of the supra-crestal mucosal attachment or biologic width (BW) may last 6 to 12 weeks [7]”. 

The word “peri-implant biological width”, commonly used to describe the apico-coronal variable dimensions of the supracrestal attached tissues is no longer accepted in the literature and should be replaced by “supracrestal tissue attachment” (Jepsen S, Caton JG, et al. Periodontal manifestations of systemic diseases and developmental and acquired conditions: Consensus report of workgroup 3 of the 2017 World Workshop on the Classification of Periodontal and Peri-Implant Diseases and Conditions. J Periodontol. 2018;89(Suppl 1):S237–S248. 

The reviewer suggests that the word "supracrestal tissue attachment " be used throughout the manuscript.

67-89 “. Classical histological studies have described the arrangement of connective tissue fibers around implants in dogs and humans…

Surface modified prosthetic abutments have shown different results from those described by the authors, creating a more robust perpendicular collagen fiber attachment to the abutment (Nevins et al., 2010; Nevins, Camelo, Nevins, Schupbach, & Kim, 2012; Shapoff, Babushkin, & Wohl, 2016, etc. ). The reviewer suggests that the authors report, albeit summarily, results of over-mentioned studies

Discussion

345-346 The term “circular ligament” is misleading to the reader since no physical attachment is described. 

Reviewer 2 Report

This manuscript reports an animal studies testing the impact of the geometry of the healing abutment to its cerstal attachment size and orientation. It is a very interesting concept to have concave healing abutment. Some comments as follows:

1. Please discuss what would happen the healing abutment is removed when taking the impression and also if the final restoration should have such emergence profile?

2. The crestal cone stability is also of great interest. Is it possible to measure and report the crestal bone level as well?

3. The final conclusion is too long...

Good

Reviewer 3 Report

Page1, the title is too long and divided into two parts.

We think "A Proof of Principle Animal Experimental Study" is just a phrase, it is suggested to change the writing here.

Page 1, the number of keywords is currently 8, and it is suggested to reduce the number.

One of the keywords is "animal study", which is too broad to be suitable and should be refined.

Page2, line 51

The authors wrote "In fact, it is well known that implant osseointegration without mucointegration is not sufficient per se, as a poor quality and quantity of peri-implant soft tissues are associated with the implant clinical failure”

The author is requested to check whether the statement here is too absolute.

It is not the same as the osseointegration concept originally proposed by Branemark et al.

It is suggested that more literatures be cited for discussion here.

Page 3, 2.2 part

Authors used "Two swine (sus scrofa), aged on average three years old".

Although the authors mentioned subsequently that the number of implant samples was 9, considering the individual differences between different swine, we suggest authors to add one more sample, where three replicates between groups are necessary in scientific studies.

Page3, line102

The authors mentioned about the experimental grouping "Test one presented a 1.5 mm concave area above the implant platform, whereas the Control abutment had a parallel-walled healing screw”

However, it is not explained why 1.5 mm concave area above the implant platform is chosen instead of other shape goods length. It is suggested to add an explanation here.

Page 4, line 137, "at P1-P2 extraction sites",

This abbreviation refers to unknown, and it is suggested to be improved.

Page5, line 172,

Authors mentioned "Animals were sacrificed 12 weeks after surgery."

However, in clinical practice, the second stage of implant surgery is often selected after 3 or 4 months. Please cite the literature here or explain the reasons for selecting 12 weeks.

The part of animal experiment method is suggested to clarify the logic and rewrite.

At present, the content under different subheadings overlaps, which may cause difficulties in understanding.

Page9,

In Figure5B and D, it can be observed that there is circular sections that were not covered by green. Please confirm whether it was just the problem of picture processing or the omission in statistical analysis?

Page10,

In Table1, authors mentioned “Data are shown as mean ± standard error of the mean (SEM).”

However, the actual table does not show data according to the above content, please check.

Page11, 3.5 Part,

Some authors concluded that "Control samples their incidence was significantly higher than in Test samples (Table 3)."

However, it seems that this conclusion cannot be drawn only from Figures-6a-c. From the picture, there are more "L" symbols marked in Figures-6A and B.

Please check and confirm the pictures and conclusions in this part.

The author mentioned in this article that Due to ethical reasons, it was decided to perform a proof-of-principle experimental study to see if Transmission Electron microscopy (TEM) could  be a suitable technique for an in depth study of the structure and distribution of collagen fibers and bundles in the  peri-implant soft tissues. 

The purpose of this experiment was only to explore the detection efficiency of TEM, but in addition to HE staining, no other mechanical detection methods were set up for comparison.

In addition, the animal experiment samples of the author are too small, and no mechanism research has been conducted.

It is suggested that the author check whether the current research results correspond to the contents in the Conclusion section, and add more in-depth discussion in the Discussion section to strengthen the logic.

none

Reviewer 4 Report

The manuscript described Electron microscopy ultra-structural analysis of collagen fibers as a good method to characterize the peri-implant soft tissues. The manuscript was clearly written. The experimental results indicated that the concave design of the abutment showed a higher percentage of area coverage by collagen accompanied by an increase in the collagen bundles size and a decrease in the longitudinally oriented bundles. The analysis was clear and convincible. It provides a good method for others to follow. However, I do not have enough background to tell if similar methods have been described before.

Round 2

Reviewer 1 Report

The authors have made the requested changes and the revised manuscript can be accepted for publication

Reviewer 3 Report

Figure3 lack scale. Please check and modify.

In Figure 5, we could still observe transverse collagen fibers which were not covered by green color, please check the quantitative analysis method and the results are accurate.

We also advised authors to add curve plots of the two groups respectively, according to their bundle size, to observe whether their distributions are accord with normal distribution.

Although authors have mentioned limitations of this study in the Conclusions section, we still hold the view that the small number of animals and implants might bring uncertainty and risk to the research results.

In the discussion section, the author mentioned other pig animal experimental studies. Based on the lack of enough sample of animal experiments in this paper, it is suggested that the author should increase the discussion of conclusions on other animal models.

"Swine" and "Pig" are both written in the full text, and it is suggested that they should be unified

“猪”å’Œ“猪”都是全文写的,建议统一
